# "Never Learned to Love Properly": A Qualitative Study Exploring Romantic Relationship Experiences in Adult Children of Narcissistic Parents

Minna Lyons [1,*] , Gayle Brewer [2] , Anna-Maria Hartley [2] and Victoria Blinkhorn [1]

1 School of Psychology, Faculty of Health, Liverpool John Moores University, Liverpool L3 3AF, UK
2 School of Psychology, University of Liverpool, Liverpool L69 3BX, UK
* Correspondence: m.t.lyons@ljmu.ac.uk

**Abstract:** Narcissism is a personality trait characterised by selfishness, coldness, entitlement, and grandiosity. There has been much research on different parenting dimensions and their relationship to narcissism in grown-up children, with a notable lack of studies investigating the influence of narcissistic parents on their children. This study focused on individuals' experiences in romantic relationships, using personal narratives from a popular 'Reddit' community for people who perceived to have grown up with narcissistic parents. Using an inductive thematic analysis on 77 Reddit posts, we identified four themes: (i) Strategies and emotions in current relationships, (ii) behaviours and characteristics of partners and their families, (iii) parent intrusiveness in current relationships, and (iv) journey to realisation and recovery. Themes are discussed in relation to existing literature and theory. We add to the sparse literature on narcissistic parents' influence in adult relationships, highlighting the importance of process from parental behaviour to adult romantic relationships.

**Keywords:** narcissistic parents; adult relationships; romantic relationships; childhood experiences

## 1. Introduction

"They f\*\*k you up, your mum and dad.
    They may not mean to, but they do.
They fill you with the faults they had
    And add some extra, just for you".

Philip Larkin (1974)

The late English poet Philip Larkin summarised in "This be the verse" what multitudes of psychology researchers have confirmed in empirical studies, parenting exerts a powerful influence on the child, which can last for the duration of their lifetime. Parents are especially influential in the success of their adult children's romantic relationships via the development of attachment styles. For example, controlling and overly critical parenting guides infant and child attachment, which can later lead to relationship insecurity (Rice et al. 2005), jealousy (Choe et al. 2021), and suboptimal emotion-regulation strategies in romantic conflict (Girme et al. 2020). One relevant yet under-researched aspect of parental influence is parental narcissism. Despite the implications for psychological therapies (e.g., Shaw 2010) and a variety of self-help guides for adult children of narcissistic parents (e.g., Harrison and Dixon 2019), surprisingly little is known about how narcissistic parents impact the romantic relationships of their adult children.

Narcissism is a socially aversive personality trait that has been a subject of much academic debate (e.g., Krizan and Herlache 2018). Researchers have proposed several models to explain what narcissism is, with grandiosity (i.e., self-centered, entitled, dominant, arrogant) emerging as a convincing model. There has been a host of quantitative studies investigating how grandiose narcissism relates to difficulties in interpersonal relationships

(e.g., Foster and Brunell 2018), which could be due to hostility, criticism, control, and coerciveness associated with the trait (Blinkhorn et al. 2016; Peterson and DeHart 2014).

Most studies on narcissism in families have focused on how parenting dimensions relate to the development of narcissism in both children (e.g., Brummelman et al. 2015) and adults (e.g., Green et al. 2020; Huxley and Bizumic 2017; Jonason et al. 2014; Lyons et al. 2013). However, there is a notable lack of research investigating the influence of narcissistic parents on their children. Current, albeit sparse, literature alludes to narcissistic individuals engaging in dysfunctional parenting styles. Studies have found that narcissistic parents may employ sub-optimal parenting strategies in terms of permissiveness and authoritarianism (Hart et al. 2017), prefer partners who are over-controlling as parents (Lyons et al. 2020) and struggle to adjust during the transition to parenthood (Talmon et al. 2020). However, very little is known about the damage that narcissistic parents can cause to their children later in life.

The few studies on this topic suggest that being brought up by a narcissistic parent can result in multiple adversities in later life. Parental narcissism relates to depression and anxiety in young adults, potentially via parenting styles characterised by high control and low care (Dentale et al. 2015). A qualitative study discovered stories of incompetence (i.e., constantly critiquing the child), isolation (e.g., denying celebrations, hobbies, and friendships), and denied childhood (i.e., neglecting and scorning the child; Määttä and Uusiautti (2020). These kinds of experiences are likely to have a long-term impact on the developing child, influencing and guiding their strategies in romantic relationships as an adult.

Indeed, relationship skills may be subject to intergenerational transfer (Kamp Dush et al. 2018), potentially via social learning. Children's perceptions of their parents' relationships may relate to the expectations that they have about romantic relationships later in life (Einav 2014). There is some empirical evidence for poor marital quality and various problems in relationships where one or both partners have high levels of narcissism (Gewirtz-Meydan and Finzi-Dottan 2018; Lavner et al. 2016). Thus, the poor relationship quality of narcissistic parents could transfer into relationship problems in adult children.

Although there are no studies looking at parental narcissism and the adoption of dysfunctional relationship models, it is clear that parent marital quality also has an association with child marital quality. It is possible that parental relationships influence the child's expectations about long-term relationships, which could relate to the child's romantic relationship quality (Zagefka et al. 2021). In turn, a positive family environment and competent parenting could harness the child with tools that are useful in relationships (e.g., conflict resolution; Xia et al. 2018). To our knowledge, no research has investigated how narcissistic parents affect the romantic relationship strategies of their adult children. One qualitative study found that those who thought their parents were poor role models (which would be expected for narcissistic parents) engaged in trial and error while seeking love, affirmation, and support that they did not get from their parents (Jamison and Lo 2021).

As well as the childhood influence, narcissistic parents can have an impact on the present lives and relationships of their adult children. Although there seems to be a cultural script around what families should be like (e.g., close and unconditionally loving; Scharp and McLaren 2017), many relations between parents and adult children are discordant, riddled with conflict (Van Gaalen and Dykstra 2006), and sometimes lead to physical or emotional estrangement (Blake 2017). There is currently very little information on how parental narcissism plays out in the current relationships of adult children and how the children may try to mitigate the harm caused by their parents.

In this research, we aim to add to the sparse literature on narcissistic parents' influence on adult relationships by employing a qualitative approach. More specifically, we analyse posts from online discussion forums for people who grew up with parents whom they perceived as narcissistic. Online support groups have been used successfully in previous studies investigating sensitive topics around families, such as experiences of adult children of alcoholic parents (Haverfield and Theiss 2014) and parental regrets (Moore

and Abetz 2019). Here, we employed a popular platform, Reddit, with a broad research question in mind: "What are the romantic relationship experiences of adult children of narcissistic parents"?

## 2. Method

### 2.1. Selection of Forum Posts

For the present study, we utilised Reddit, a discussion forum platform with over 10,000 user-generated "subreddits", anonymous online communities that are unified by common interests (Widman 2020). The anonymity provides an opportunity for users to share sensitive or stigmatising information in a manner that may be less threatening than face-to-face discussion (Sowles et al. 2017). Indeed, the empowering, social, and supportive aspects of these online communities have previously been acknowledged (van Uden-Kraan et al. 2009). Reddit has been successfully used to research topics, such as partner violence (Lyons and Brewer 2021), mental health (De Choudhury and De 2014), and parenting (Sutter et al. 2021). For the present study, we identified one subreddit with a large number of users (at the point of data collection, over 691,000 users). This subreddit is specific to individuals who identify as being raised by narcissistic parents.

We selected initial posts and responses to posts that discussed personal experiences in romantic relationships of adult children of narcissistic parents. Only posts that had the focus on the users' own romantic relationships were chosen (rather than those that mentioned relationships in passing while concentrating the discussion on another topic). We searched the subreddit by using a combination of search words, "romantic, relationships, dating, marriage". Posts that discussed the experiences of someone else or gave advice without sharing own experiences were excluded. Overall, 77 subreddit posts were deemed appropriate. The data collection was concluded at the point where no further relevant posts could be found with the aforementioned search terms. The username, link to the post, forum user age and gender, identity of the narcissistic parent (mother, father, both), current status of the parent–child relationship (in-contact or no-contact), and country of origin of the posts were recorded (where possible). Age was provided in 56 posts, with a mean age of 23 years (range 16–50 years). Posts were typically written by women ($n = 60$), with relatively few by men ($n = 13$) or those not identifying their gender ($n = 4$). Forum posts described relationships with a narcissistic mother ($n = 34$), narcissistic mother and father ($n = 26$), or narcissistic father ($n = 14$). For three posts, the gender of the narcissistic parent was not clear. Approximately half of the forum users ($n = 37$) indicated that they were in contact with the narcissistic parent, 12 reported no contact, and 23 did not reveal this information. The origin of many forum posts was not clear ($n = 28$), however, there were posts written in the U.S.A. ($n = 31$), Canada ($n = 6$), India ($n = 4$) and South Africa ($n = 2$) and one each from Chile, China, France, Hungary, Japan, and Korea. All the posts were written in 2020 and 2021.

### 2.2. Ethical Issues

Though the posts were publicly available, ethical approval was obtained from the Institutional Review Board (ref: 9754). Further, we consulted relevant ethical guidelines, previously published discussion forum research, and available guides to discussion forum research (e.g., Smedley and Coulson 2021) when designing, conducting the study, and reporting our findings. We analysed posts available to the general public without registration or login and adopted a number of measures in accordance with professional body guidelines (e.g., British Psychological Society 2017) in order to protect the anonymity of the forum users. We are not revealing the name of the subreddit or online usernames and have slightly altered the wording of the quotes in this report. To further address this issue, we entered each quote into both Google (the most widely used search engine) and Reddit (the discussion forum platform used to obtain posts) to ensure they did not lead to the original posts.

*2.3. Data Analysis*

We employed inductive, reflexive thematic analysis as the analytical strategy for the data (e.g., Braun and Clarke 2021). The reflexive approach is characterised by allowing the coding to aid the theme development, and the theme construction takes place relatively late in the analytical process.

The data were extracted manually from a popular Reddit site by two researchers (A.-M.H. and M.L.). After saving the data on a word file, one researcher (M.L.) read and re-read through the posts several times, developing initial codes. The whole research team read several of the posts prior to an online meeting where the codes were discussed. As a result, some of the codes were removed, and others were amalgamated together. M.L. led the development of themes from the agreed codes, and the themes were discussed with all the team members. The team members consisted of three academics with robust experience in narcissism research, as well as an undergraduate student intern who has an interest in the topic. Moreover, some of the research team members had their own experiences with narcissistic parents, which was useful in terms of discussing and understanding the themes that we extracted from the data.

The original data cannot be accessed due to confidentiality issues. We do not want any information to be released which could compromise the identity of the participants and their online profiles.

**3. Results**

We constructed five themes from the codes: (i) Strategies and emotions in current relationships, (ii) behaviours and characteristics of partners and their families, (iii) parent intrusiveness in current relationships and (iv) journey to realisation and recovery. In total, the 47 quotes in the results were drawn from 36 different individuals. When the information is available, the quotes are accompanied by the gender and age of the poster, and the gender of the parent.

(i)    Strategies and Emotions in Current Relationships (61% of Posts)

Forum users discussed several relationship strategies and feelings within relationships, many of which were seemingly maladaptive in nature. Often, they made direct links between these behaviours and their childhood experiences (e.g., "I never stay single for long. I'm either cold/distant or dedicated to a sometimes-unhealthy degree. I realise this is me trying to replicate the love I was anxious to receive from nparents", post 30, gender and age unknown, both parents narcissistic). In other posts, people discussed communication difficulties, stemming from lack of models for healthy relationships in childhood (e.g., "I have terrible communication issues. Never learnt how to love properly. I give partners absolutely everything, and they don't give it back. It keeps me invested", post 35, gender and age unknown, parent gender unknown). Users discussed issues, such as fear of abandonment/exploitation, commitment difficulties/fear of intimacy, feelings of guilt/shame/anger, need for validation, lack of conflict resolution strategies, overt sensitivity to criticism, trust issues, and lack of boundaries.

For example, one issue that was raised was a lack of trust in other people, something that they had learned from the narcissistic parent. One individual discussed how her mother " … told me that men bring problems in life, and she is the only one to rely on … As a consequence, although I wish to have romantic relationships, I don't think of them as a forever thing. I always thought relationships are just for one or two years" (post 7, female, age 32, narcissistic mother). Many felt like their parents damaged their ability to have healthy romantic relationships, partially due to an inability to resolve conflicts, "I feel damaged all the time, with intense feelings of guilt and shame if I do something wrong. Even littlest fights turn into something big, and more emotional than they should be" (post 12, gender and age unknown, both parents narcissistic). It was evident that many of the posters were hypersensitive to any negative emotions from their partner.

One of the issues around lack of trust was the fear that the current partner would turn out to be just like the narcissistic parent. This was something that clearly undermined the quality of the relationships and made the individual unnecessarily suspicious. For instance, "If I see any similarities between my dad and boyfriend, I get scared. If my boyfriend says something similar to what my dad has said I think it's a red flag. I feel that I turn things into red flags that are not necessarily red flags" (post 74, female, age unknown, narcissistic father).

Many talked about feelings and behaviour related to extreme commitment anxiety. Commitment to long-term relationships was difficult because of fear of being loved. For example, "I was head over heels planning our future, felt like we have known each other all of our lives. After we both confessed love, I just freaked. I just completely fell out of love in a matter of days" (post 44, male, age unknown, gender of parent unknown).

Commitment anxiety and fear of intimacy seem to stem from childhood experiences with narcissistic parents. Often, posters mentioned that their parents were over-controlling and condemning. There was a fear that if the person showed their real self to their romantic partner, they would make themselves vulnerable to critique and exploitation. Some discussed how they had learned how love is short-term and conditional. For example, in the words of one poster, "Mother always made a point of how I needed her much more than she needed me. I think this is the reason why I'm so distant now with romantic partners. I go into relationships assuming that the other person will eventually find faults with me, and abandon me" (post 16, gender and age unknown, narcissistic mother).

The posters also lacked an understanding of boundaries in relationships. They discussed how they had learned to put other people first and felt like they had a sole responsibility for their partners' happiness. One person stated that "My childhood relationship with a narcissistic mother means that I do not understand the importance of people respecting my boundaries. I always thought I have to put other people first. In previous serious relationships, people constantly disrespect my boundaries, and I let them do it. I prioritised their feelings, wants, and interests over mine. I allowed myself to feel responsible for their happiness. It was draining" (post 18, gender and age unknown, narcissistic mother). Mistreatment by partners was often allowed because the person thought that this kind of behaviour was normal. The models of interpersonal relationships acquired in childhood meant that the person often ended up being treated badly by their partner. "I did not know what it is like to feel happy, respected, and have my feelings taken into account. I put up with whatever because I did not know how to respect myself or know my worth. I let people treat me badly because I thought it is normal" (post 54, gender and age unknown, gender of parents unknown).

(ii)   Behaviours and Characteristics of Partners and their Families (47% of Posts)

This theme featured posts that discussed what kind of characteristics the partners possessed (e.g., kind/abusive), and the dynamics between the parents, partners, and the partners' families. The posters' relationships with the parents were a cause of conflict with the partners or in-laws. For example, the decision to have no contact with the parents was often met with suspicion by the in-laws. Indeed, the estrangement from parents was perceived as a red flag to partners and/or their families, "People who are not close with their families are referred to as red flags. Although people want to relate to you, they can't. I had an ex who said to me once that I hate everyone's family because you don't have one" (post 11, female, age unknown, gender of parents unknown).

For some, their partners' families tried to mediate contact with the narcissistic parent. The posters were left feeling like the parent is the victim, and cutting contact with them was an overreaction from the adult child. One user talked about their mother-in-law's complete lack of understanding of the no-contact situation, blaming it on the adult child's immaturity and over-sensitivity, "She's trying to mediate between me and the parents. The worst is that she is telling them information (address, phone number, pictures, pregnancy) about me. I have asked her to stop, telling her about the verbal and physical abuse, controlling

behaviour, and my depression. She cannot comprehend why I would take such drastic measures" (post 40, female, age 30, both parents narcissistic).

Other posters were worried that their partners would not want to stay with them because of their narcissistic parents. They were fearful that the parents' behaviour would result in relationship dissolution, for example, "My boyfriend (of 2 years) and I planned to get engaged. After seeing my parents for thanksgiving, he is convinced that marriage merges the families, and wants nothing to do with them" (post 68, female, age unknown, both parents narcissistic).

Many also outlined the positive characteristics of their partners. They praised them for being kind, loving, and understanding, for example, "I am so lucky and grateful to have married an amazing man who is nothing but supportive" (post 60, female, age unknown, gender of parents unknown). The partners were often fundamental to defence against the narcissistic parents, especially if they had similar experiences with their own parents, "My partner is defensive and protective of me, not afraid of calling out my mom when she gaslights, controls, or agonizes me. He has narcissistic parents and is hyper-aware of these tactics" (post 6, female, age unknown, narcissistic mother).

Conversely, some wrote about how their partners had the characteristics of narcissistic parents, "It is not possible for me to have a relationship with "normal" people because my brain is hardwired by two narcissistic parents. I cannot relate to anyone normal, and will always attract more narcs, borderlines, and sociopaths" (post 17, gender and age unknown, both parents narcissistic). One poster discussed how the emotional abuse in childhood has made them attracted to abusive individuals in adult romantic relationships, "I've had a couple of very narcissistic ex-girlfriends. I wonder if I have some sort of Stockholm Syndrome, making me naturally attracted to them" (post 22, male, age 28, both parents narcissistic). The posters sometimes drew direct relationships between the negative partner characteristics and their own upbringing. Self-hate, compounded with fear of loving and being loved, seemed to draw individuals to emotionally abusive partners.

The narcissistic parents also had a large influence on the lives of the partners, causing mental distress and relationship difficulties. One person wrote how "my mother always tried to convince me that who I'm with is abusive. My husband of 10 years has tried hard to show her how good he is. My mother is telling everybody how bad he is treating us. It really hurts my husband, and he is depressed" (post 29, female, age unknown, narcissistic mother). It was obvious that having to deal with the narcissistic in-laws was a distressing experience to the partners, often causing additional strain in the relationship.

(iii) Parent Intrusiveness in Current Relationships (31% of Posts)

This theme includes the emotions (e.g., jealousy, resentfulness, need to compete, fear of abandonment) and behaviours (e.g., manipulation, control, harassment) that narcissistic parents employ when their adult children are in relationships. One of the common features in the posts was around control. The parents had an excessive need to control the adult child's relationships, including restricting contact with the partner. For example, "My mother tries to control our relationship from the sidelines, only allowing us to see each other once or twice a month...not allowing me to meet his family, saying that "you are not getting married tomorrow... she is just being insanely protective. It comes from a fear of losing control" (post 4, female, age 21, narcissistic mother). Narcissistic parents had the need to exert control over their child's lives, failing to accept that the child has grown into an autonomous adult.

Many individuals discussed how the parents were using strategies for manipulation and harassment, including trying to turn the partner against the child. One of the aims of the manipulation seemed to be the wish to break up the relationship. One poster described how "My narcissistic mother tried to turn my partner against me by telling him I sleep around with dirty men, that I'm crazy, don't shower, anything negative you can think of" (post 26, female, age 26, narcissistic mother). The same poster wrote how the manipulation strategies even included attempted seduction of the partner, "She even tried to proposition my ex-husband for sex". For some, the parents were successful in

breaking up the relationship, "I dated him for over 2 years, and every time I went to see him, my mother would scream at me. The abuse about the relationship got so bad I just had to end it. I was so traumatized and emotionally exhausted I just wanted it to end. The minute I broke up with him, she acted completely different towards me" (post 34, female, age unknown, narcissistic mother). One parent suggested that the daughter should "...cheat on my boyfriend, even though I don't want to! It's like she's projecting hatred from her own marriage and wants me to feel the same" (post 69, female, age unknown, narcissistic mother).

Narcissistic parents were resentful and jealous of their child's relationships. For example, one individual wrote how "narcissistic mothers never want your relationships to thrive. Mine had a complete breakdown when I was getting married. She reacted to my engagement by saying, "you guys are not thinking this through". She was negative throughout the planning of the wedding, didn't even come to shop for a dress" (post 51, female, age unknown, narcissistic mother). One daughter mentioned how her mother " . . . actually said, "congratulations, you win" to my (now) fiancé, like I was some kind of a prize to be fought over" (post 67, female, age unknown, narcissistic mother). It seemed like many parents were in competition with the partner for the attention of their child.

(iv) Journey to Realisation and Recovery (60% of Posts)

The final theme was about change, how the person came to the realisation that their parents do not define who they are, how they took active steps to change their situation (e.g., therapy, no-contact with parent), how the partners were instrumental in recovery, and how the online community helped them in the journey.

Many shared how their partners were important in the process of recovery. For instance, one poster described how she had been " . . . dating a guy for almost two years and it has been the best time of my life! It's just the way he treats me. Growing up with a narcissistic mother, I became this needy, anxious, apologetic mess. He helped me see it is not my fault, and that it's ok to be myself" (post 77, female, age unknown, narcissistic mother). Another person discussed how "My partner was instrumental in healing. They made me communicate my needs and taught me healthy boundaries. They make me feel loved in a way I never thought would be possible. They opened my eyes, and I can never thank them enough" (post 53, age and gender unknown, gender of parents unknown).

The Reddit community had an important role in the journey to recovery. The posters praised the community for understanding the issues that the individual was facing and for giving advice on how to cope. One poster talked about how "This subreddit is amazing . . . Reading and relating to so many posts has been of tremendous aid in helping me to navigate my feelings of stress and anxiety. I have realised that my behaviours in interpersonal relationships stem from the fact that I had no emotional support when growing up" (post 4, female, age 23, narcissistic mother). Knowing that one is not alone was a powerful experience, "I realise that although sometimes I feel crazy because I have been raised by narcissistic parents, and it's such a lonely experience, there are millions out there who have experienced the same. I'm not alone or crazy" (post 10, male, age unknown, both parents narcissistic).

Posters commonly talked about "realisation", or "epiphany" around having narcissistic parents, and how their experiences no longer defined who they were. One person wrote that " . . . I realised that it is ridiculous to convince others that my childhood abuse and resulting PTSD is not that bad... Because it was that bad, it wasn't my fault, and my narcissistic family does not define me" (post 27, female, age unknown, both parents narcissistic). Another disclosed that "I had an epiphany when I realised that I wasn't just worthy of love, but I was a goddamn catch! I was nice, cute, enthusiastic in bed, smart, loyal, good cook, and financially independent. I could and should be picky. I should find someone worthy of me" (post 48, female, age unknown, gender of parents unknown). One individual talked about improving themselves, and how they have " . . . seen a lot of progress. As an adult, it doesn't matter what has happened in the past. It's about what you are going to do to fix it. I'm open to anything that helps me to become a better person.

I want to be able to accept and give love in a way that is healthy" (post 73, male, age unknown, narcissistic mother).

Therapy was also a common experience in realisation and recovery. One person wrote how "Upon therapy and further analysis, I came to realise that I associate family with entrapment, living to serve others, not pursuing your dreams, depression . . . Marriage and family mean prison, and all the negative emotions that go with that" (post 7, female, age 32, narcissistic mother). Others had not yet had therapy but thought that it would be beneficial, "CBT might help, but I feel I need some serious neurological remodelling in this brain of mine. I feel like my parents have broken me, but I'm determined not to let their unhappiness tarnish my future" (post 35, age and gender unknown, gender of parents unknown).

Many spoke about how they had cut ties with their parents and how the no-contact relationship helped them to recover from the abuse. For instance, one person explained how no-contact had improved her marriage, "I haven't had a single fight with my husband since going no-contact. I feel we are closer, it's so chill, and we have been lavishing lots of love on our toddler. Highly recommend" (post 63, female, age unknown, narcissistic father). Another individual discussed how they were " . . . raised in a cult-like environment with overbearing and opinionated narcissistic parents who did a lot of damage. I'm in no-contact now . . . and that's my hope and strength, in addition to my faith" (post 17, gender and age unknown, both parents narcissistic).

## 4. Discussion

The current study focused on experiences in romantic relationships, using personal narratives from a popular Reddit community for people who grew up with narcissistic parents. Using an inductive thematic analysis, we constructed themes around emotions and relationship strategies in current relationships (which often directly stemmed from childhood), the behaviours and characteristics of partners and their families, behaviours of narcissistic parents around the adult child's current relationship, and positive journeys to realisation and recovery. The findings highlight the importance of process, from parental behaviour to adult romantic relationships (see also findings in Jamison and Lo 2021).

The theme around strategies and emotions in current relationships suggested that the parenting received in childhood had a direct influence on romantic relationships in adulthood. The Reddit community shared childhood stories of excessive parental control, criticism, physical and emotional abuse, manipulation, love withdrawal, and dysfunctional relationship models. As adults, childhood experiences were related to low trust, feelings of shame, commitment difficulties, and poor relationship strategies. Many of the current experiences have links to adult attachment theories. The mental models of self and others acquired from caregivers in childhood are applied in adult romantic relationships, leading to secure or insecure attachment styles (e.g., Hazan and Shaver 1987). Individuals in our sample recalled their narcissistic parents as cold and controlling. These parenting dimensions have been related to the development of adult relationship attachment styles, characterised by avoidance (e.g., avoiding commitment, hiding feelings) and anxiety (e.g., excessive worry about being abandoned, neediness; Díez et al. 2019). Thus, the parenting of a narcissistic caregiver may influence romantic relationships in adulthood via insecure attachment styles.

As well as being subjected to abusive and neglectful parenting, the individuals wrote about how they lacked any models of "normal" relationships. There is evidence of intergenerational transmission of hostile relationships via social learning (Kamp Dush et al. 2018; Masarik and Rogers 2020), which could be what we saw in our sample. Many discussed how their conflict resolution strategies had not developed because of the poor parental models, leading to unnecessary escalation of arguments. These findings mirror quantitative studies that have found that children in high-conflict families may lack the skills to resolve conflicts in their romantic relationships later in life (Xia et al. 2018). Overall, the Redditors

in the study were well aware that the relationship models from their childhood continued to influence their adult romantic relationships.

The second theme was centered around characteristics, behaviours, and feelings of the partner (which often was directly linked to narcissistic parents), as well as in-laws and their opinions about the relationship. For instance, the posters discussed their partner's characteristics both in positive (e.g., unlike their parent, kind, supportive, loving, understanding) and negative (e.g., just like their parent, narcissistic, manipulative, abusive) light. These opposing characterisations of the partners contradict the psychoanalytical and evolutionary theories asserting that individuals seek their parents' personality in their romantic partners (see also McCrae et al. 2012). Partner choice is a complex process, and parents' personality is unlikely to be a strong factor in interpersonal attraction in adulthood (Gyuris et al. 2010).

In addition to the characteristics of the partners, the posters often talked about how their partners or in-laws failed to understand why they had cut contact with their parents. This suggests that when the cultural script of close and unconditionally loving families (e.g., Scharp and McLaren 2017) is broken, the person initiating no-contact is viewed as a red flag. Adult children who are estranged from their parents often experience stigma from others and are subject to negative perceptions (i.e., being selfish or childish; Rittenour et al. 2018). The Redditors in our study were no exception to the stigma of estrangement from parents (see also Blake 2017).

The third theme highlighted an under-investigated, under-theorised area of research: The dynamics of relationships between parents and their adult children (Kirby and Hoang 2018). Many of the parents used intrusive, manipulative tactics in trying to control their children to comply with the wants and needs of the parent. Empirical research has linked intrusive parenting to poorer romantic relationship quality in young adults (Parise et al. 2017), which could have origins in insecure attachment and relationship jealousy (Choe et al. 2021). It was obvious that the individuals in the current study not only had to deal with the psychological consequences of intrusive parenting in childhood but that the intrusiveness carried on in current romantic relationships too. There is a desperate need for more studies investigating the relationship dynamics between narcissistic parents and their adult children. Indeed, to date, the whole field of research on adult children and their parents has been largely neglected in psychology (Kirby and Hoang 2018).

The fourth theme consisted of posts containing a wealth of positive messages in terms of a realisation that the parent was an abusive narcissist. This realisation might have important clinical implications, as understanding abuse in terms of abuser narcissism could be beneficial for recovery (Marsden et al. 2021). In a similar way to the qualitative findings in Määttä et al. (2020), the posts evidenced resilience, recovery, and an attempt to improve romantic relationships. More than half (60%) of the posts discussed how the person was actively seeking or receiving help (e.g., from the partners, Reddit community, therapy). For many, estrangement from the parent provided a space to breathe and reflect. A range of healthy recovery tools can provide a platform for effective recovery from narcissistic parents (see also Edery 2019), and the individuals in our sample drew on a wealth of resources.

The Reddit community was especially instrumental in giving and receiving valuable peer support. The value of online communities has been demonstrated for a range of topics, from domestic violence (e.g., Lyons and Brewer 2021) and adult children of alcoholic parents (Haverfield and Theiss 2014) to sexual violence victimisation (O'Neill 2018). The act of sharing online with peers who have similar experiences can result in disclosing information a person has not talked about before (O'Neill 2018). Overall, the posts were characterised by overwhelmingly positive experiences with mutual sharing in the online community. Peer-led therapeutic approaches have been effective in recovery from the trauma of abuse (Konya et al. 2020) and are certainly something that should be explored further in the adult children of narcissistic parents.

*Limitations and Future Directions*

Although our study provides initial qualitative evidence for narcissistic parents' influence on adult children's romantic relationships, this study has limitations. First, we cannot verify whether the parents were, indeed, narcissistic. It is possible that some of the posters were inaccurate in assessing their parents' narcissism. However, familiarity and intimacy with a person seem to increase the accuracy of personality evaluations (Connelly and Ones 2010), which suggests that adult children should be the experts in judging their parents' personality. In addition, especially grandiose narcissism seems to be detected in others with high accuracy (Miller et al. 2011), even with minimal information (e.g., interview transcriptions, Brunell et al. 2021). Thus, we do think that the Redditors in our study have experienced parenting from narcissistic caregivers.

Second, online forums are a form of self-reported information provided by users, and as such, it is impossible to know whether the content of the posts is accurate. However, the anonymity afforded by the Reddit platform provides an opportunity to share sensitive or stigmatising information in a more comfortable manner than in offline interactions (Sowles et al. 2017). Indeed, it is widely accepted that when anonymity is guaranteed, individuals are more likely to be honest with the information they provide (e.g., Link and Mokdad 2005).

Third, due to the lack of information concerning the background details of the sample, we often could not verify the location, gender, age, or other demographic details. It is possible that variables such as gender of the narcissistic parent intersect with the gender of the child (see also Umberson 1992). Future studies could use mixed methods in mapping the interaction between parent and child variables to the types of relationship difficulties the child might be experiencing. Parents seem to have a sex-specific effect on their children (e.g., empathy development; Lyons et al. 2017), and this could be taken into consideration in future studies.

## 5. Conclusions

In summary, this study focused on experiences in romantic relationships, using personal narratives from a popular Reddit community for people who grew up with narcissistic parents. We have added to the sparse literature around narcissistic parents' influence on the adult relationships of their children and found a number of important themes which highlight the importance of process, from parental behaviour, to adult romantic relationships. Further research is encouraged to broaden our current limited knowledge within this important aspect of childhood to adulthood.

**Author Contributions:** Conceptualization, M.L., G.B., V.B. and A.-M.H.; methodology, M.L., G.B., V.B. and A.-M.H.; formal analysis, M.L. and A.-M.H.; investigation, M.L., G.B., V.B. and A.-M.H.; data curation, M.L. and A.-M.H.; writing—original draft preparation, M.L.; writing—review and editing, M.L., G.B. and V.B. All authors have read and agreed to the published version of the manuscript.

**Funding:** This research received no external funding.

**Institutional Review Board Statement:** The study was conducted in accordance with the Declaration of Helsinki, and approved by the Institutional Review Board of the University of Liverpool (protocol code 9754, February 2021).

**Informed Consent Statement:** Participant consent was waived due to open online nature of the data and impracticality of seeking for consent in this context.

**Data Availability Statement:** We will not publish the data due to the need to further protect the online community.

**Acknowledgments:** We would like to thank all the people who contributed to the discussions around this topic. Many friends/family/colleagues/ people with lived experiences contributed to the making of this paper by sharing their knowledge and ideas. It would be impossible to name them all. Special thanks go to Noora Hietanen and our dear late friend/sister Nina Tuomi who unfortunately did not

live to see this paper published. We would also like to thank all Redditors—we hope that we have represented your experiences in a sensitive and thoughtful manner.

**Conflicts of Interest:** The authors declare no conflict of interest.

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
