# Peer review of "“Never Learned to Love Properly”: A Qualitative Study Exploring Romantic Relationship Experiences in Adult Children of Narcissistic Parents"

_socsci, doi:10.3390/socsci12030159_

Round 1

Reviewer 1 Report

The studies qualitatively explored how the romantic relationships of Reddit users raised by narcissistic parents are influenced by their upbringing. The study was certainly intriguing, and the manuscript held my interest throughout.

Some things to be mindful of: there were about 3 spelling mistakes/grammar that I caught and there may be more that I didn’t catch – Line 430 “origings” should be “origins”, Line 200 “unnecessary” should be “unnecessarily”, and Line 82 “impact” should be “an impact”.

The authors give specific examples of posts represented by each theme, but it would be helpful to know how many or the percentage of each post represented by each theme.

The discussion claim of intergenerational transmission of hostile relationships is a bit far reaching, given that we don’t really know how many posts actually reflected the theme of hostility.

Author Response

Many thanks for reviewing our paper! We have now corrected the typos. After a thorough proof-reading, we also found several other typographical errors, which are now corrected too.

We don't think it is necessary to indicate the percentage that each post represented in the theme. This would be very time-consuming to calculate, as the individual posts were snippets of larger posts from the individuals. We hope that knowing what percentage of the total posts were represented in each theme will be sufficient information.

We have now highlighted that intergenerational transmission is via social learning. There is such a plethora of evidence to suggest that relationship styles can be learned from parents that we think this suggestion is plausible. 

Again, many thanks for taking your time to read our work, we really appreciate it!

Reviewer 2 Report

This is an interesting and well-written paper addressing a current gap in adult children's experiences of narcissistic parents and how they believe they affect their romantic relationships. 

There are a few small typos and the manuscript should be read for final editing - several missing apostrophes, check for consistency in italics in quotes, and some capitals ('Redditors' also sometimes written as 'redditors').

As a next step, it would be interesting to include the views of other people - therapists, parents, partners? It could also be helpful to discuss if there is any difference to the type of people using reddit, compared to others? 

Author Response

This is an interesting and well-written paper addressing a current gap in adult children's experiences of narcissistic parents and how they believe they affect their romantic relationships. 

We would like to thank the reviewer for taking their time to comment on our paper! Thank you for spotting the typos and inconsistencies- all of these should now be corrected. 

We love the suggestions for next steps- comparison between reddit and other populations would be really interesting. Also, the views of the grandchildren would be worthy of investigation. Thank you for giving us more food for thought!